# Distinctiveness of Femoral and Acetabular Mesenchymal Stem and Progenitor Populations in Patients with Primary and Secondary Hip Osteoarthritis Due to Developmental Dysplasia

**DOI:** 10.3390/ijms25105173

**Published:** 2024-05-09

**Authors:** Mihovil Plečko, Nataša Kovačić, Danka Grčević, Alan Šućur, Andreja Vukasović Barišić, Tea Duvančić, Ivan Bohaček, Domagoj Delimar

**Affiliations:** 1Department of Orthopaedic Surgery, University Hospital Center Zagreb, 10000 Zagreb, Croatia; mplecko@kbc-zagreb.hr (M.P.);; 2Laboratory for Molecular Immunology, Croatian Institute for Brain Research, School of Medicine, University of Zagreb, 10000 Zagreb, Croatia; 3Department of Anatomy, School of Medicine, University of Zagreb, 10000 Zagreb, Croatia; 4Department of Physiology and Immunology, School of Medicine, University of Zagreb, 10000 Zagreb, Croatia; 5General Hospital “Dr. Anđelko Višić” Bjelovar, 43000 Bjelovar, Croatia; 6Department of Innovative Diagnostics, Srebrnjak Children’s Hospital, 10000 Zagreb, Croatia; tea.duvancic@gmail.com; 7Department of Orthopaedic Surgery, School of Medicine, University of Zagreb, 10000 Zagreb, Croatia

**Keywords:** osteoarthritis, hip, developmental dysplasia of the hip, mesenchymal stem cells

## Abstract

Primary hip osteoarthritis (pOA) develops without an apparent underlying reason, whereas secondary osteoarthritis arises due to a known cause, such as developmental dysplasia of the hips (DDH-OA). DDH-OA patients undergo total hip arthroplasty at a much younger age than pOA patients (50.58 vs. 65 years in this study). Recently, mesenchymal stem and progenitor cells (MSPCs) have been investigated for the treatment of osteoarthritis due to their immunomodulatory and regenerative potential. This study identified cells in subchondral bone expressing common MSPC markers (CD10, CD73, CD140b, CD146, CD164, CD271, GD2, PDPN) in vivo and compared the proportions of these populations in pOA vs. DDH-OA, further correlating them with clinical, demographic, and morphological characteristics. The differences in subchondral morphology and proportions of non-hematopoietic cells expressing MSPC markers were noted depending on OA type and skeletal location. Bone sclerosis was more prominent in the pOA acetabulum (Ac) in comparison to the DDH-OA Ac and in the pOA Ac compared to the pOA femoral head (Fh). Immunophenotyping indicated diagnosis-specific differences, such as a higher proportion of CD164+ cells and their subsets in DDH-OA, while pOA contained a significantly higher proportion of CD10+ and GD2+ cells and subsets, with CD271+ being marginally higher. Location-specific differences showed that CD271+ cells were more abundant in the Fh compared to the Ac in DDH-OA patients. Furthermore, immunohistochemical characterization of stromal bone-adjacent cells expressing MSPC markers (CD10, CD164, CD271, GD2) in the Ac and Fh compartments was performed. This research proved that immunophenotype profiles and morphological changes are both location- and disease-specific. Furthermore, it provided potentially effective targets for therapeutic strategies. Future research should analyze the differentiation potential of subsets identified in this study. After proper characterization, they can be selectively targeted, thus enhancing personalized medicine approaches in joint disease management.

## 1. Introduction

Osteoarthritis (OA) is a progressive degenerative joint disease whose pathogenesis is still not completely understood [1,2]. It is characterized by cartilage degradation, osteophyte and cyst formation, inflammation of the synovium, and subchondral sclerosis [3]. Primary OA develops without an apparent underlying reason [1]. However, it is not just a simple “wear and tear” process but a disease that affects the joint as an organ, with many factors influencing its development [4]. Risk factors associated with primary OA are age, obesity, genetic predisposition, mechanical and alignment factors, prior joint trauma, etc. [4,5]. Recent reports highlight that subchondral bone plays a vital role in the pathogenesis of OA [3]. Also, some consider OA an inflammatory disease because several cytokines play a role in its pathophysiology [6]. Studies are usually performed on knee OA, and the results are then extrapolated to hip OA patients [7]. However, primary hip OA (pOA) seems to differ from knee OA in pathophysiology, epigenetics, anatomy, prevalence, etc. [7]. Therefore, data obtained from research on other joints should be extrapolated with caution. When OA arises due to a known cause, such as developmental dysplasia of the hips (DDH), it is called secondary OA [8]. DDH is associated with numerous genes, which may explain the incidence variability in different regions of the world [9]. 

Various methods of OA management are described, such as physical therapy, adapted occupational therapy, pharmacotherapy, food supplementation, intra-articular injections, and surgical management [6,10]. A study showed that the mean age when female patients suffering from pOA undergo total hip arthroplasty (THA), considered a treatment of last resort, is 71.6 years, and 69.1 years in men [11]. Crowe et al. classified secondary hip OA caused by developmental dysplasia of the hips (DDH-OA) into four grades based on the degree of femoral head dislocation on radiographs [12,13]. Zhen et al. showed a group of DDH-OA Crowe grade 1 and 2 patients whose mean age was 32.5 years when they had THA [14].

In 2007, THA was named the “operation of the century” [15]. A study showed that a THA would last for 15 years in around 88% of patients, and for 25 years in around 58% of patients, respectively [16]. The number of people over the age of 60 in the world is increasing, along with the increase in life expectancy [17]. Therefore, the development of biological methods for the treatment of OA is gaining importance [15,18]. Currently, there is no possibility of articular cartilage regeneration or slowing down the development of OA [2]. Biological methods of treating cartilage damage include joint injections of platelet-rich plasma and growth factors, bone marrow stimulation techniques such as microfractures of the subchondral bone, osteochondral graft transplantation, autologous chondrocyte implantation, matrix-associated chondrocyte implantation, and cell-based therapy such as application of bone marrow aspirate or stromal vascular fraction from fat tissue into joints [19]. However, by using most of these, it is still not possible to achieve the formation of high-quality articular cartilage, while other procedures that provide high-quality cartilage are considered complex procedures with limited indications [2]. In recent years, mesenchymal stem and progenitor cells (MSPCs) from various sources, i.e., bone marrow, have been investigated for the treatment of OA due to their immunomodulatory and regenerative potential [20]. Furthermore, MSPCs are also known as medicinal signaling cells, which better describes their ability to act at the site of injury or disease, where by secreting bioactive factors, they behave as a medicinal substance in situ [21]. The International Society for Cell and Gene Therapy issued minimum criteria for cells to be considered MSPCs [22]. Churchman et al. showed that native bone marrow-derived in vivo MSPC populations can perform various functions and may be adequate targets for OA treatment in situ [23]. However, the MSPC population is heterogeneous, and in order to identify subpopulations possessing better regenerative capacity, various cell markers were investigated [24]. Churchman et al. state that MSPCs found in vivo are significantly different from their in vitro cultured MSPC counterparts depending on their topographic niche, highlighting the need for performing studies of MSPCs in OA in vivo [23]. Still, there is a paucity of studies regarding in vivo present progenitor cells in OA, especially in hip OA. One such study is by Rasini et al., describing bone marrow cells expressing MSPC markers to be present in vivo [25]. They further divided them into four subsets based on the co-expression of CD10, CD73, CD140b, CD146, GD2, and CD271 [25]. Another study is by Chan et al., who recently described a human skeletal stem cell (hSSC) defined as the PDPN+CD146−CD73+CD164+ subset, which reportedly may differentiate into bone, cartilage, and stroma but not adipose tissue [26]. Furthermore, they used those markers, together with CD90, to define subsets by their differentiation potential and lineage commitment (chondrogenic vs. osteogenic/stromal) [26]. The hSSCs were further identified in vivo in samples obtained from pOA patients [26].

The aim of the present study was to identify cells in subchondral bone expressing common MSPC markers in vivo and to compare the proportions of these populations in end-stage pOA vs. DDH-OA, as well as to correlate them with the clinical, demographic, and morphological characteristics of the two groups. Identification of MSPC subsets that are more frequent in pOA and DDH-OA provides great targets for analysis of differentiation potential and thus identification of possible future targets for biological treatment of OA.

## 2. Results

### 2.1. Patient Cohort

Demographic and clinical characteristics of patients suffering from pOA and DDH-OA are shown in Table 1. The DDH-OA group was significantly younger, while no significant difference was found comparing gender distribution, body mass index (BMI), or duration and intensity of symptoms.

### 2.2. Morphological Characteristics of Femoral and Acetabular Samples

The main morphological inter-group difference was observed between acetabulum (Ac) samples, with the pOA Ac having significantly greater total bone area/tissue area (BA/TA) vs. the DDH-OA Ac, while subchondral and trabecular BA/TA were greater; however, the differences were not statistically significant (Figure 1). Femoral head (Fh) samples were seemingly without significant morphological differences.

Intra-group analysis showed BA/TA and subchondral BA/TA were significantly greater in the Ac compared to the Fh in the pOA group, while in the DDH-OA group, there were no evident morphological differences (Table 2).

### 2.3. Composition of Femoral and Acetabular Populations

Analysis of cells extracted from Ac and Fh samples by digestion showed that the median number of live cells per sample in the DDH-OA group was 78,597.5 [25,568.5–859,500] compared to 70,967 [16,726–153,797] in the pOA group, *p* = 0.235. The median proportion of non-hematopoietic stromal cells (CD31−CD45−CD202b−CD235a−) in the DDH-OA group was 0.81 [0.56–0.99] %, while in the pOA group, it was 0.78 [0.38–1.48] %, *p* = 0.844. The location-specific comparison showed that Ac samples had a median of 106,136 [29,383.5–394,454.5] live cells, while Fh samples had 41,800 [18,188.5–162,778] live cells, *p* = 0.190. The median proportion of stromal cells in Ac samples was 0.70 [0.4–1] % vs. Fh samples 0.85 [0.6–1.2] %, *p* = 0.160.

### 2.4. The Disease-Specific Phenotype of Femoral and Acetabular Single-Positive Mesenchymal Stem and Progenitor Cells between pOA and DDH-OA Groups

For detailed phenotype characterization and quantification of MSPC subpopulations, two flow-cytometry panels were applied on paired Ac and Fh samples and compared between pOA and DDH-OA groups. Differences in MSPC composition between pOA and DDH-OA were more prominent in Ac than in the Fh subchondral bone compartment (Figure 2). In the Ac samples, the CD164+ subpopulation was significantly enlarged in the DDH-OA group, while in the pOA group, the CD10+ subpopulation was significantly enlarged and the GD2+ subpopulation was enlarged with marginal significance (*p* = 0.0519). In Fh samples, the GD2+ subpopulation was significantly enlarged in the pOA group.

### 2.5. The Site-Specific Phenotype of Single-Positive Mesenchymal Stem and Progenitor Cells within pOA and DDH-OA Groups

In the pOA group, no significant differences in composition were observed for subpopulations defined by the expression of a single marker; however, the GD2+ subpopulation seems to be enlarged in Ac samples, with marginal significance (*p* = 0.0640) (Table 3).

In the DDH-OA group, a difference was noted in the CD271+ subpopulation, which was significantly enlarged in Fh compared to Ac (Table 4).

### 2.6. Immunohistochemical Characterization of Cells Expressing Mesenchymal Stem and Progenitor Markers in pOA and DDH-OA

To further confirm the presence of MSPCs in the subchondral compartments of the pOA and DDH-OA samples, we performed in situ localization of marker expression (Figure 3 and Figure 4).

Ac samples from patients with DDH-OA had an enlarged population of CD164+ non-hematopoietic cells in comparison to samples from patients with pOA according to flow cytometry. Immunohistochemical analysis showed a positive CD164 signal of high intensity in different regions of the Ac (Figure 5). The most apparent signal was in the bone marrow, coming from small round cells. Furthermore, a positive signal was noted on larger round cells in the marrow. Other positive signals were noted from round cells in the perivascular area, bone marrow stroma, and Haversian canals, i.e.*,* endosteum. Also, spindle-shaped cells with a positive signal were noted in the bone-lining area and perivascular area, as well as in the stroma adjacent to the bony tissue.

Analysis of Fh samples showed a similar distribution of cells with a positive signal for CD164 (Figure 6).

Analysis of CD10 in Ac samples showed a positive signal on small and large round cells in the bone marrow. Also, a positive signal was noted on the fibroblastoid-like cells in the stroma. Analysis of Fh samples further showed positive signal in ring-like cells that seem to be of adipose tissue origin.

The GD2 signal was mostly positive in small round cells of the bone marrow in both Ac and Fh samples. Also, a positive signal was observed in spindle-shaped cells in the bone marrow stroma, as well as in the bone-lining area and the perivascular area.

Small round cells in the bone marrow seem to express CD271 both in Ac and Fh samples. Furthermore, a positive signal for CD271 was also noted in spindle-shaped cells in the bone-lining area, stroma adjacent to bone, and the perivascular area.

Additional imaging is provided in Appendix A.

### 2.7. Subpopulations of Cells Defined by Multiple Mesenchymal Stem and Progenitor Markers in pOA and DDH-OA

According to previous studies indicating that MSPC subpopulations of different differentiation potential should be defined using multiparameter phenotyping, we further expanded our analysis to multiple MSPC markers and stratified in vivo present MSPCs. From our first panel, we observed that in DDH-OA patients, the CD164+CD146− subpopulation was significantly larger in the Ac compared to the pOA Ac (Figure 7). Although total CD73 positive cells did not differ between pOA and DDH-OA, the proportion of CD164+CD146− cells co-expressing the CD73 marker was significantly higher in the DDH-OA Ac in comparison to pOA. Although CD164+CD146−CD73+PDPN+, i.e., hSSC, were detected in our samples, they did not differ between groups. Subpopulations in Fh samples did not significantly differ between groups.

From our second panel, we observed that the pOA Ac samples had enlargement of the CD10+GD2+, CD271+GD2+, and CD271+GD2+CD10+ subpopulations in comparison to the DDH-OA Ac (Figure 8). A subpopulation of CD10+CD271+ was detected; however, it did not differ between groups. Also, subpopulations in the Fh samples did not significantly differ between groups.

Site-specific analysis showed a significantly larger subpopulation of CD164+CD146−PDPN+, as well as CD10+GD2+, CD271+GD2+ and CD271+GD2+CD10+ subpopulations in the pOA Ac compared to the pOA Fh (Table 5). In the DDH-OA group, no site-specific difference in subpopulations was observed (Table 6).

## 3. Discussion

This study demonstrated the differences in subchondral morphology and proportions of non-hematopoietic cells expressing distinct MSPC markers depending on OA type and skeletal location. Bone sclerosis was more prominent in the pOA Ac in comparison to the DDH-OA Ac and in the pOA Ac compared to the pOA Fh. In parallel, we demonstrated the presence of stromal bone-adjacent cells expressing MSPC markers in the Ac and Fh compartments. Immunophenotyping indicated diagnosis-specific differences, such as an overall higher proportion of non-hematopoietic CD164+ cells and their subsets in DDH-OA, while pOA samples contained a significantly higher proportion of CD10+ and GD2+ cells, with CD271+ being marginally higher. Moreover, the triple-positive subpopulation (CD10+GD2+CD271+) was significantly enlarged in pOA. Location-specific differences showed that non-hematopoietic CD271+ cells were more abundant in the Fh compared to the Ac in DDH-OA patients. Furthermore, the CD10+GD2+, CD271+GD2+, CD271+GD2+CD10+, and CD164+CD146−PDPN+ subsets were enlarged in the pOA Ac compared to the Fh.

One of the main hallmarks of OA is subchondral bone sclerosis, together with cartilage degradation [1]. The pOA Ac had a significantly higher proportion of BA/TA compared to the pOA Fh. Cartilage is thicker in the peripheral part of the healthy Ac, while in the healthy Fh, cartilage is thickest in the central area [27,28]. Therefore, the difference in BA/TA might be due to the anatomical location from where we obtained samples. Also, the pOA Ac has a significantly higher proportion of BA/TA compared to the DDH-OA Ac. In DDH-OA, due to anatomical changes, there is an asymmetric distribution of forces in the hip [13,29,30]. It might be that the DDH-OA Ac samples were obtained from an anatomically similar location, but not a biomechanically similar location, as the pOA Ac samples were. A study comparing Ac MSPCs and Fh MSPCs of pOA patients, as well as our study, found significant differences between these populations, advocating location-specific composition of MSPCs [31]. Therefore, these morphological changes also may be due to a possible different pathogenesis of OA in the Ac compared to the Fh or perhaps a “shift in the time points” in various locations of the progression of the disease itself. However, further research is needed to confirm either of the theories.

To the best of our knowledge, this is the first study characterizing the in vivo MSPC populations in DDH-OA and pOA patients, highlighting the difference in pathogenesis and regenerative capacity between the two groups. Čamernik et al. compared Fh samples of patients with pOA and DDH-OA and found, similar to our study, an equal proportion of CD45/CD19/CD14/CD34-negative MSPCs in both groups, but contrary to our results, a reduced population of live cells was found in patients with pOA [32]. Furthermore, MSPCs obtained from pOA samples had lower chondrogenic and osteogenic potential in vitro [32].

The main finding of our study is a higher proportion of CD164+ and CD164+CD146−subpopulations in DDH-OA patients compared to pOA, which is especially pronounced in the Ac. Also, an enlarged subpopulation of CD164+CD146−CD73+ cells was noted in the DDH-OA Ac compared to pOA. CD164+ cells have a known role in early hematopoiesis [33]. On the other hand, the literature on CD164+ as a marker of MSPCs is scarce. Jasenc et al. found a significantly lower proportion of in vitro expanded bone marrow-derived CD164+ MSPCs from Fh trabecular bone in early pOA compared to late pOA and patients without OA [34]. They have further proposed that depletion of CD164+ cells may serve as a marker of pOA onset [34]. As mentioned earlier, Chan et al. described hSSC as PDPN+CD146−CD73+CD164+, which reportedly may differentiate towards the osteochondral lineage but not adipose tissue [26]. We have identified this subpopulation in both patient groups; however, we did not find a significant difference in the proportion between groups. Immunohistochemical analysis showed CD164+ round cells located in the bone marrow, in the perivascular area, bone marrow stroma, and Haversian canals, i.e.**,** endosteum. Also, CD164+ spindle-shaped cells were found in the bone-lining area, perivascular area, and the stroma adjacent to the bone. Most of these cells probably belong to the hematopoietic lineage; however, this study showed that a significant proportion of stromal cells in bone express CD164, and further research is warranted to clarify the role and potential of these cells as MSPCs.

In the pOA group, CD10, GD2, and CD271 seem to be expressed in subpopulations that are enlarged compared to DDH-OA, mostly in Ac samples. Xu et al. report on CD10+ cells in the tunica adventitia of blood vessels that have stronger osteogenic potential and promote bone formation in vivo [35]. Ding et al. noted that CD10+ adventitial cells exhibited higher proliferation and were clonogenic and osteogenic potentials in comparison to their CD10− counterparts, playing a role in perivascular MSPC function [36]. Moreover, Graneli et al. showed that MSPCs increase the expression of CD10+ if differentiated towards the osteogenic and adipogenic lineages [37]. Animal studies report that GD2+ bone marrow MSPCs are more committed to differentiating to osteoblasts and adipocytes but showed enhanced expression of pluripotency markers (SSEA-1, Nanog) [38,39]. Martinez et al. report that within bone marrow, only MSPCs expressed GD2+, advocating that GD2+ is the first single-surface marker of MSPCs [40]. Rasini et al. reported that CD271+ and CD10+ markers are expressed in round stromal cells located in the bone marrow stroma and medullary cavity, and these cells express pluripotency markers (Oct4, Nanog, SSEA-4) [25]. Furthermore, CD271+GD2+ are expressed on bone-lining cells located in the endosteum and also express pluripotency markers (Oct4, SSEA-4) [25]. CD10+GD2+ are expressed on fibroblastoid reticular cells located in bone marrow stroma, the medullary cavity, and perivascular areas; however, they do not express pluripotency markers [25]. Our study found CD10, GD2, and CD271 to be expressed on round cells located in the bone marrow, indicating some of them might belong to the round stromal cells or fibroblastoid reticular cells earlier described by Rasini et al. [25]. Also, CD10 was expressed on ring-like cells, which might be adipose stromal cells described by the same group of authors, which express pluripotency markers as well (Oct4, Nanog). Furthermore, we found GD2 and CD271 expressed on spindle-shaped cells in the endosteum, which may correspond to bone-lining cells [25]. Moreover, we have also observed cells in the perivascular area that express GD2 and CD271. Taking into consideration that the BA/TA value was significantly higher in the pOA Ac vs. the DDH-OA Ac, these findings may suggest that pOA samples are more osteogenically committed compared to DDH-OA.

Location-specific analysis showed that the pOA Ac is more sclerotic compared to the Fh, especially in the subchondral area. Furthermore, the pOA Ac also had a significantly larger proportion of CD10+GD2+, CD271+GD2+, and CD271+GD2+CD10+ cells. As described earlier, it seems these markers represent cell populations that are committed toward the osteogenic lineage. In DDH-OA, a CD271+ population was significantly enlarged in Fh compared to Ac samples. However, the differences in bone area between locations in DDH-OA were not significant. Barilani et al. showed that CD271+ adult MSPCs have high clonogenic and osteogenic properties compared to CD271− cells [41]. Jones et al. report that CD271+ MSPCs are abundant in the trabecular bone niche and indistinguishable from CD271+ MSPCs aspirated from bone marrow in terms of their osteogenic commitment [42]. Ilas et al. describe the accumulation of CD271+ MSPCs of predominantly osteogenic potential in the sclerotic regions of the Fh in pOA patients [43]. Sivasubramaniyan et al. reported on different subpopulations of CD271+ cells in the Fh localized in the bone-lining regions and the perivascular area [44]. Campbell et al. analyzed the Fh of pOA patients by magnetic resonance, comparing areas with bone marrow lesions and those without bone marrow lesions [45]. They found that CD271+ MSPCs significantly accumulate in the lesion areas [45]. Nguyen et al. compared cultures of MSPCs from the subchondral bone marrow of the Ac and Fh obtained during THA with cultures of MSPCs obtained from bone marrow aspirates [46]. They showed that MSPCs from the Ac have a greater ability to form colonies, while MSPCs from the Fh have a more pronounced osteogenic potential compared to MSPCs from the Ac [46]. Trivanović et al. also found a higher proportion of CD271+ MSPCs in the pOA Fh compared to the Ac [31]. Moreover, Ac MSPCs showed osteogenic and chondrogenic potential, but their adipogenic capacity was lower compared to the Fh MSPCs [31]. Kuci et al. showed that CD271+ cells have a higher degree of adipogenic capacity compared to non-selected MSPCs [47]. Trivanović et al. hypothesize that the Ac and Fh in pOA host distinct mesenchymal cell phenotypes [31]. Barilani et al. state that the CD271+ subpopulation seems to be heterogeneous, confirming the need for more specific markers to define MSPCs and their properties [41]. We believe that a more detailed identification of CD271+ MSPCs is indeed needed, as there might be a difference when considering the location of the sampling and function of these cells.

Limitations of this study include that flow cytometry and histological analysis were not performed from the same sample but from adjacent samples. As OA is a “geographical” disease, this has to be taken into consideration when interpreting the results. Histomorphometric characterization should be interpreted with caution, as it does not account for mineralization quantity or quality. Moreover, subchondral and trabecular BA/TA morphometry was performed by choosing a randomly allocated square from the middle third of the sample as the most intact part of the sample. However, this area is not representative and these results need to be interpreted with caution. Nevertheless, the results did not lead to any major conclusions made in this study. Furthermore, immunohistochemical analysis was performed without double staining for distinct hematopoietic markers (namely CD45). As most of the markers are also positive on hematopoietic cells, one has to be careful in the interpretation of data gained from the immunohistochemical analysis. Moreover, one of the limitations is the number of patients per group, which is relatively small. Therefore, we could not address with certainty whether age, BMI, or other demographic data influence any of the noted changes. Also, due to ethical reasons, it is not possible to compare our results to results from healthy age-matched donors.

## 4. Materials and Methods

### 4.1. Patients

A total of 24 consecutive patients with pOA (N = 12) and DDH-OA (N = 12) admitted to the Department of Orthopedic Surgery between February 2021 and April 2022 and scheduled for THA were included in the study. The study was approved by the institution’s Ethics Committee under Class 8.1-20/201-2 No. 02/21 AG. All patients signed the informed consent document. All procedures were carried out according to the Declaration of Helsinki. The inclusion criteria for the pOA group were age 18–80 and the diagnosis of pOA grade 3 or 4 by Kellgren–Lawrence radiographic classification of OA severity [48,49]. The inclusion criteria for the DDH-OA group were age 18–80, Kellgren–Lawrence grade 3 or 4, and the diagnosis of DDH-OA, Crowe grade 1 or 2 (Appendix A) Exclusion criteria for both groups were a current or past inflammatory joint disease, earlier surgeries of the hip, prior or existing malignancies, or systemic diseases affecting the musculoskeletal system. All patients were requested to fill out a general information question form, modified Harris Hip Score questionnaire, and WOMAC questionnaire to assess their pain, stiffness, and physical function.

### 4.2. Sampling

Osteochondral samples were harvested from the Ac (two samples) and the Fh (two samples) right after joint exposure, prior to performing a THA, from tissue considered surgical waste. The sampling was performed with a 10 mm diameter cylindrical chisel (Small Joint OATS Set, 10 mm AR-8981-10S, Arthrex, Naples, FL, USA) and a sample height of 5 mm. (Appendix A) According to Wasielewski et al., Ac is divided into 4 quadrants which differ regarding the distance of neurovascular structures [50]. Boundaries of the quadrants were determined intraoperatively using two imaginary lines: a line goes through the spina iliaca anterior superior and passes through the center of Ac, while the second line crosses the first line at 90° at the center of Ac. Both samples were taken from the superior posterior quadrant, which represents the location of the greatest distance from the neurovascular structures. Ilizaliturri et al. divided Fh into 6 zones, defined by two vertical (part of Fh in contact with anterior and posterior limits of acetabular fossa, respectively) and one horizontal imaginary line (perpendicular to vertical lines, crossing the part of Fh in contact with superior limit of acetabular fossa) [51]. Two samples were taken from the central upper and rear upper zones, corresponding to the area of contact with the area on Ac from which the samples were taken. All of the samples were further processed within 6 h of surgery: one of each Ac and Fh was stored in fresh saline solution (0.9% NaCl) and the other in 10% formaldehyde (BioGnost, Zagreb, Croatia).

### 4.3. Histology

Samples were stored in 10% formaldehyde (BioGnost) for 4–6 days at room temperature, and the solution was exchanged daily. Decalcification was performed using a solution of 1.5% HCl (Sigma-Aldrich, Darmstadt, Germany) and 5% formic acid (Sigma-Aldrich) in distilled water, exchanged daily from 15 to 34 days at RT, depending on the sample, with daily incubations at 60 °C for 2 h. After decalcification, the samples were washed under water for 24 h and then left for an additional 24 h in 4% formaldehyde. They were finally dehydrated, cleared, and embedded in paraffin. Paraffin blocks were cut on a rotary microtome (Leica, Nussloch, Germany) into 5 μm thick sections and mounted onto positively charged Superfrost Plus slides (Menzel-Gläser, Thermo Scientific, Schwerte, Germany). Histological sections were stained with Goldner trichrome staining according to the manufacturer’s instructions.

The immunohistochemical analysis for CD10, CD164, CD271, and GD2 was performed on selected patients with pOA (N = 3) and DDH-OA (N = 3). Slides were deparaffinized in xylene and rehydrated in ethanol. Antigen retrieval was performed with overnight incubation in citrate buffer (pH 6.2). Inactivation was performed using hydrogen peroxide. Blocking was performed with 10% goat serum (G9023, Sigma-Aldrich, Taufkirchen, Germany). Samples were incubated with anti-CD10 (MA5-14050, 1:5, Thermo Fisher Scientific, Rockford, IL, USA), anti-CD164 (ab238748, 1:150, Abcam, UK), anti-CD271 (ab3125, 1:30, Abcam, UK), and anti-GD2 (LS-C63496, 1:30, LifeSpan BioSciences, Seattle, WA, USA) primary antibodies overnight at 4 °C, diluted in Cell Signaling Ab diluent (Cell Signaling Technology, Danvers, MA, USA). Negative control samples were incubated with 1% goat serum diluted in Cell Signaling Ab diluent. The signal was visualized using the Dako REAL EnVision Detection System (K500711-2, Agilent Technologies, Santa Clara, CA, USA) according to the manufacturer’s instructions.

Following covering and drying, the slides were examined under a microscope (CX33, Olympus, Tokyo, Japan) and photographed with the accompanying digital camera (EP50, Olympus, Tokyo, Japan).

### 4.4. Histomorphometry

Histomorphometry was performed in a blinded fashion with an Olympus CX33 (Olympus, Tokyo, Japan) microscope and a concomitant digital camera EP50 (Olympus). All samples were photographed under the same conditions. Whole samples were photographed under 40**×** magnification and compiled into a single image using Automate->Photomerge->Reposition command in Adobe Photoshop CC 2015 (Adobe Inc., San Jose, CA, USA). After image preparation, black and white image masks were generated, representing bone and soft tissue, respectively [52,53]. The color threshold was set to 128. The masks were used to quantify total BA/TA (%) (Appendix A). Furthermore, in the middle third of the sample on the image, as the structurally most preserved part, a 500 μm × 1000 μm box was randomly allocated in the subchondral area, measuring BA/TA (Subchondral BA/TA; %) in the same manner. Trabecular area (Trabecular BA/TA; %) was measured by placing another 500 μm × 1000 μm box 2000 μm deeper to the subchondral box.

### 4.5. Preparation of Single-Cell Suspension

Samples stored in saline solution were used to prepare single-cell suspensions. The cartilage was removed using a scalpel, and the sample was mechanically fragmented with a bone rongeur. The preparation was carried out according to the previously described protocol, with minor modifications [54]. Bone particles were immersed in 1 mg/mL Collagenase type IV from Clostridium histolyticum (Sigma #C-5138, Sigma-Aldrich, St. Louis, MO, USA) in 0.1 M phosphate-buffered saline (PBS) and incubated at 37 °C for 1 h. The sample was then filtered through cotton gauze to remove the remaining bone parts and resuspended using a 22 G needle. The suspension was then filtered using a 100 μm nylon cell strainer, centrifuged for 5 min at 250 g at 4 °C, and resuspended in PBS. Erythrocytes were lysed using hypotonic lysis buffer (150 mM NH_4_Cl, 1 mM KHCO_3_, and 0.1 mM Na_2_EDTA, pH 7.4).

### 4.6. Flow Cytometry

The cells (4–5 × 10^7^/tube) were first incubated with a solution for blocking human Fc-receptors (Human TruStain FcX, Biolegend, San Diego, CA, USA) for 15 min at room temperature. Phenotyping included two antibody panels to characterize MSPC subpopulations. Both panels included conjugate antibodies to human hematopoietic markers as dump channel: CD31-APC (Biolegend, Cat#303116, clone WM59, 1:100), CD45-APC (Biolegend, Cat#368512, clone 2D1, 1:100), CD202b-AF647 (Tie-2) (BioLegend, Cat#334210, 1:100), and CD235a-APC (Biolegend, Cat#306608, clone HIR2 (GA-R2), 1:100). First mesenchymal panel included: CD73-FITC (BioLegend, Cat#344016, 1:100), PDPN-PE (Thermo Fisher Scientific, Waltham, MA, USA, Cat#12-9381-42, 1:200), CD146-PECy7 (BioLegend, Cat#342010, 1:100), and CD164-Purified (Biolegend, Cat# 324802, 1:100). Second mesenchymal panel included: GD2-FITC (Biolegend, Cat#357314, 1:100), CD140b (PDGFRβ)-PE (Biolegend, Cat#323605, 18A2, 1:100), CD271 (NGFR)-PECy7 (Biolegend 3 Cat#45110, 1:100), and CD10-APCCy7 (Biolegend, Cat#312212, 1:100). The cells were incubated for 30 min at 4 °C, protected from light and rinsed with PBS. Secondary staining was performed for the first panel with goat anti-mouse IgG APCCy7 (minimal x-reactivity) antibody (BioLegend, Cat#405316, 1:200) for 25 min, following the previously described protocol [54]. Finally, each sample was resuspended in PBS and 7-amino-actinomycin D (7-AAD, BioLegend) was added to exclude dead cells. Samples were acquired by an Attune flow cytometer (Applied Biosystems, Foster City, CA, USA). Results were analyzed using the FlowJo software (FlowJo, v10, Ashland, OR, USA) according to the gating strategy presented in Appendix A. Visual presentations of cell clusters were performed by using a T-distributed stochastic neighbor embedding (tSNE) algorithm in FlowJo software (automatic learning configuration, with 1000 iterations, perplexity 30, exact KNN algorithm, and Barnes–Hut interpolation algorithm, including concatenated non-hematopoietic populations from all samples in each group and using compensated fluorescence parameters for each marker [55]). The tSNE plots show fluorescence intensity for each parameter (heatmap view).

### 4.7. Statistical Analysis

Throughout the text, for all the covariates we report either the median [interquartile range] or the arithmetic mean ± standard deviation. Statistical analysis was performed using MedCalc (version 20.006; MedCalc Software Ltd., Ostend, Belgium). Differences between the two cohorts were analyzed in the following way. For discrete data, a Chi-squared independence test was used. For continuous data, either a parametric two-sample unpaired t-test was used (when the distributions were approximately Gaussian), or a non-parametric Mann–Whitney U test for unpaired samples and non-parametric Wilcoxon test for paired samples were used (for distributions that were not approximately Gaussian). We report the *p*-values with two significant digits, and *p*-values smaller than 0.001 are reported as <0.001. All *p*-values below the level ≤ 0.05 were considered to be significant.

## 5. Conclusions

We found that the DDH-OA Ac has a significantly larger CD164+ subpopulation of MSPCs compared to the pOA Ac, further identifying the location of CD164 expression in histological sections. Also, we have identified a higher proportion of CD164+CD146− and CD164+CD146−CD73+ subpopulations in the DDH-OA Ac, which might belong to an hSSC differentiation pathway.

Our study identified that the pOA Ac and Fh have a greater proportion of CD10+ and GD2+ populations, as well as subpopulations that express CD271 compared to DDH-OA, suggesting pOA samples might be more committed towards the osteogenic and adipogenic lineages. Furthermore, bone sclerosis was more prominent in the pOA Ac compared to DDH-OA Ac.

A higher proportion of CD271+ cells in the DDH-OA Fh was noted compared to the DDH-OA Ac. On the other hand, CD10+GD2+, CD271+GD2+, CD271+GD2+CD10+, and CD164+CD146−PDPN+ subsets were enlarged in the pOA Ac compared to the pOA Fh. Also, bone sclerosis was more prominent in the pOA Ac compared to the pOA Fh.

We have proved that immunophenotype profiles and morphological changes are both location- and disease-specific. This research provides potentially effective targets for therapeutic strategies in the future. The identification of these targets is the first step towards their successful use for therapeutic purposes. However, research is warranted that will analyze the differentiation potential of subsets identified in this study. After proper characterization, they can be selectively targeted, thus enhancing personalized medicine approaches in joint disease management.

## Figures and Tables

**Figure 1 ijms-25-05173-f001:**
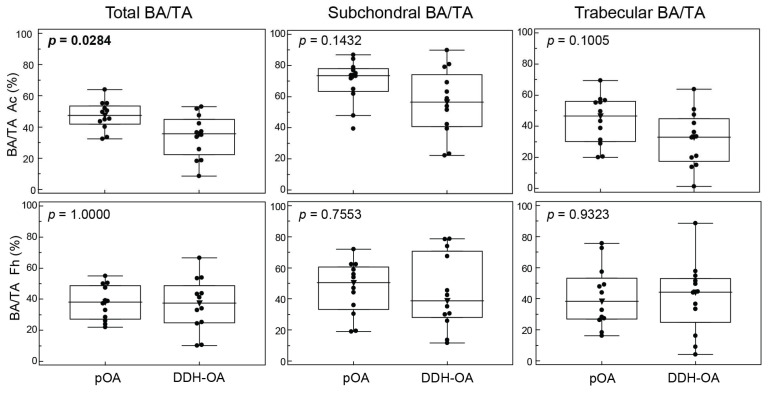
Proportions of bone area/tissue area (BA/TA) from total, subchondral, and trabecular area from acetabular (Ac) and femoral head (Fh) bone samples of patients suffering from primary hip osteoarthritis (pOA) and secondary osteoarthritis due to developmental dysplasia of the hips (DDH-OA). Horizontal lines and boxes are median and IQR; statistical significance is shown on plots (*p* < 0.05, Mann–Whitney U test). Results that were statistically significant are bolded.

**Figure 2 ijms-25-05173-f002:**
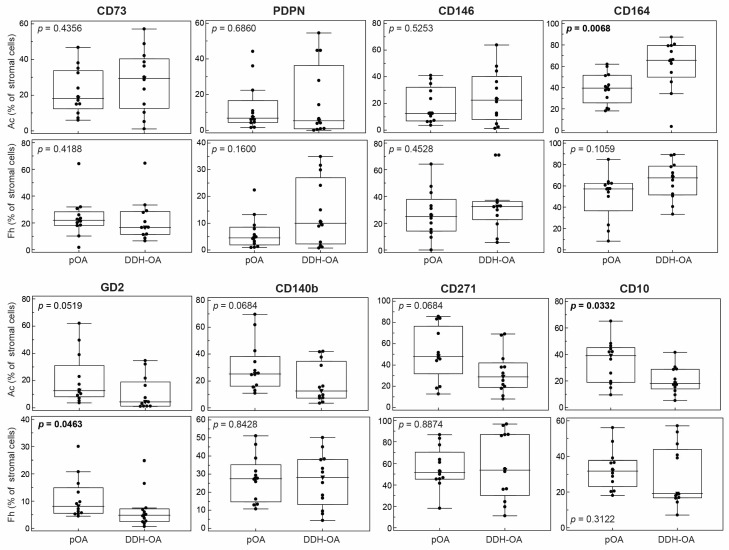
Proportions of non-hematopoietic cells positive for different mesenchymal stem and progenitor cell markers from acetabular (Ac) and femoral head (Fh) subchondral bone samples of patients suffering from primary hip osteoarthritis (pOA) and secondary osteoarthritis due to developmental dysplasia of the hips (DDH-OA). Horizontal lines and boxes are median and IQR; statistical significance is shown on plots (*p* < 0.05, Mann–Whitney U test). Results that were statistically significant are bolded.

**Figure 3 ijms-25-05173-f003:**
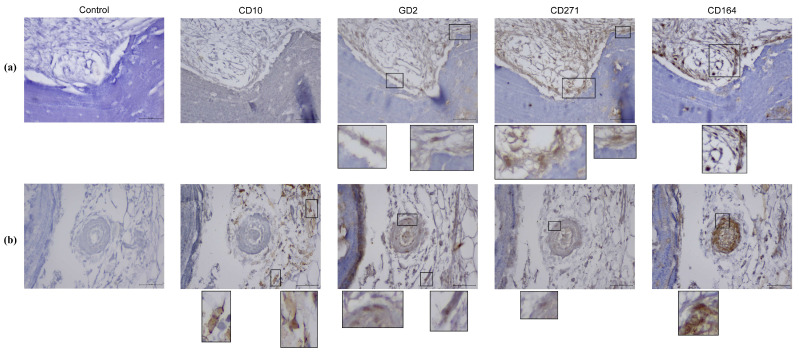
Mesenchymal stem and progenitor cell marker expression in vivo on primary hip osteoarthritis acetabulum samples. (**a**) Immunohistochemical stains show a positive signal from spindle-shaped cells in the bone marrow stroma, as well as in the bone-lining area for GD2, CD271, and CD164, and a positive signal for CD164 in small round cells of bone marrow. (**b**) Immunohistochemical stains show a positive signal on the fibroblastoid-like cells in the stroma for CD10, while a positive signal for GD2, CD271, and CD164 was noted on spindle-shaped cells in the perivascular area. Original magnification 400*×*, inset digital magnification 1200*×*. Scale set to 50 µm.

**Figure 4 ijms-25-05173-f004:**
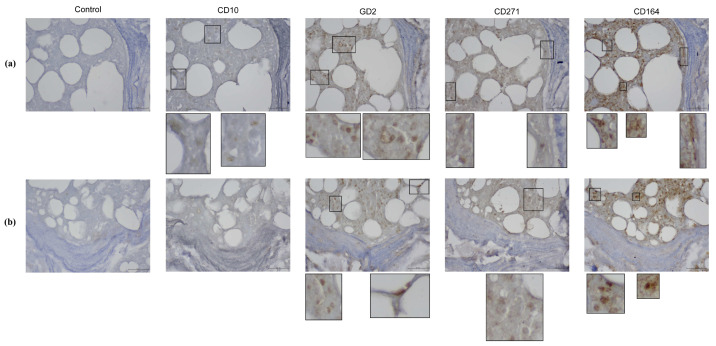
Mesenchymal stem and progenitor cell marker expression in vivo on secondary osteoarthritis due to developmental dysplasia of the hips acetabulum samples. (**a**) Immunohistochemical stains show a positive signal from small round cells in the bone marrow stroma for CD10, GD2, CD271, and CD164. Also, a positive signal was noted in the bone-lining area for CD271 and CD164. (**b**) Immunohistochemical stains show a positive signal from small round cells for GD2, CD271, and CD164. Also, a positive signal was noted from a spindle-like cell in bone marrow stroma for GD2. Original magnification 400×, inset digital magnification 1200×. Scale set to 50 µm.

**Figure 5 ijms-25-05173-f005:**
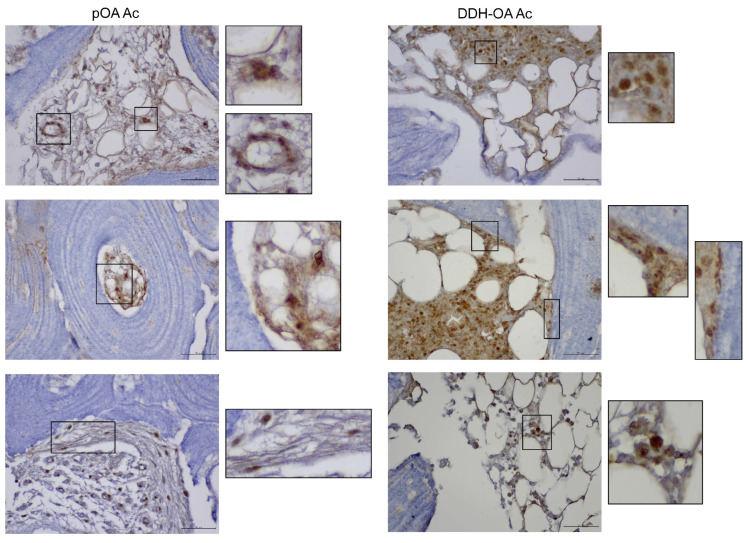
Expression of CD164 in vivo on primary hip osteoarthritis acetabulum samples (pOA Ac) and secondary osteoarthritis due to developmental dysplasia of the hips acetabulum (DDH-OA Ac) samples. A positive signal was noted from small and larger round cells in the marrow, round cells in the perivascular area, bone marrow stroma, and Haversian canals (endosteum). Also, spindle-shaped cells with a positive signal were noted in the bone-lining area and the stroma adjacent to bone. Original magnification 400×, inset digital magnification 1200×. Scale set to 50 µm.

**Figure 6 ijms-25-05173-f006:**
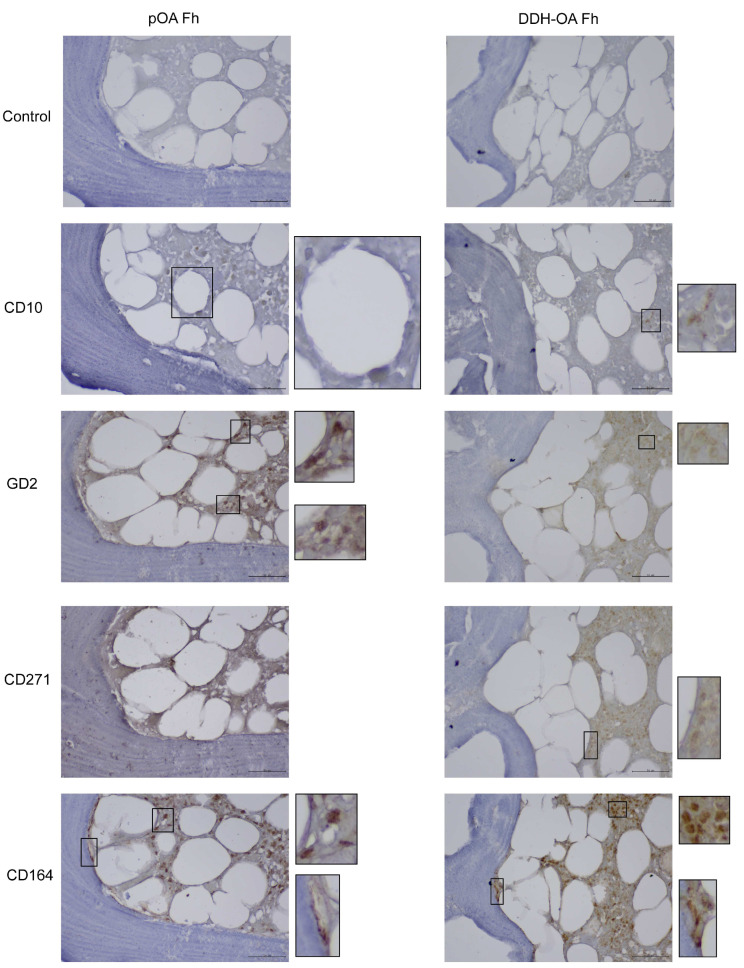
Mesenchymal stem and progenitor cell marker expression in vivo on primary hip osteoarthritis femoral head samples (pOA Fh) and secondary osteoarthritis due to developmental dysplasia of the hips femoral head (DDH-OA Fh) samples. A positive signal was noted from small round cells for CD10, GD2, CD271, and CD164. A ring-like cell was positive for CD10, while spindle-shaped cells were positive for CD164 in the bone-lining area. Original magnification 400×, inset digital magnification 1200×. Scale set to 50 µm.

**Figure 7 ijms-25-05173-f007:**
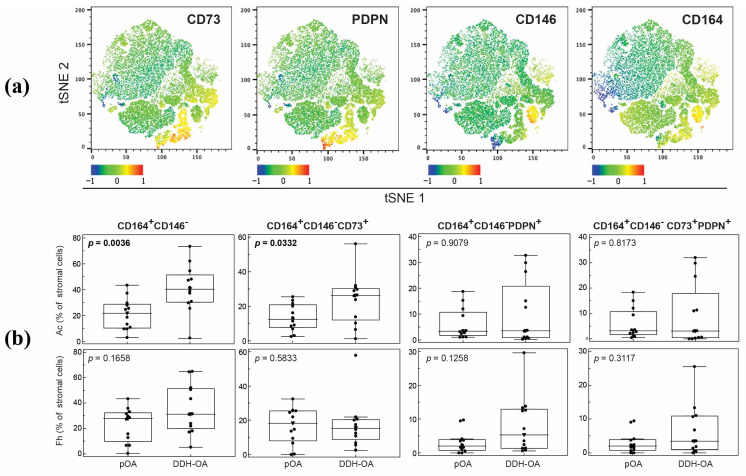
(**a**) Visual representations of non-hematopoietic cell clusters performed by T-distributed stochastic neighbor embedding (tSNE) algorithm in FlowJo software. Each marker is visualized using the heatmap statistic based on the fluorescence intensity of a compensated parameter. (**b**) Proportions of non-hematopoietic cells co-expressing different markers from acetabular (Ac) and femoral head (Fh) bone samples of patients suffering from primary hip osteoarthritis (pOA) and secondary osteoarthritis due to developmental dysplasia of the hips (DDH-OA). Horizontal lines and boxes are median and IQR; statistical significance is shown on plots (*p* < 0.05, Mann–Whitney U test). Results that were statistically significant are bolded.

**Figure 8 ijms-25-05173-f008:**
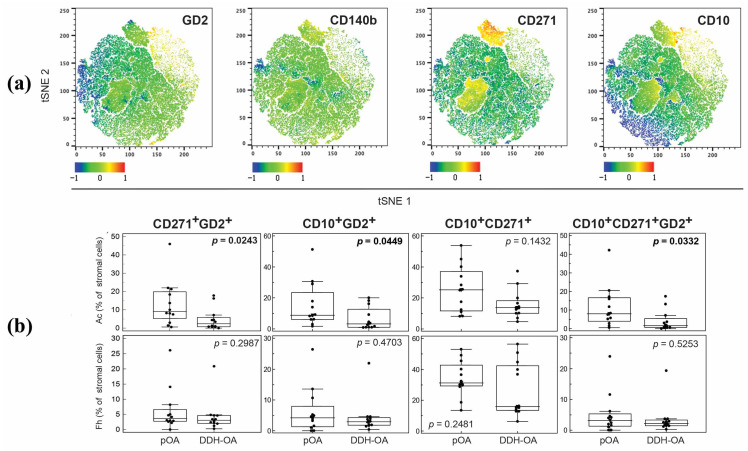
(**a**) Visual representations of non-hematopoietic cell clusters performed by T-distributed stochastic neighbor embedding (tSNE) algorithm in FlowJo software. Each marker is visualized using the heatmap statistic based on the fluorescence intensity of a compensated parameter. (**b**) Proportions of non-hematopoietic cells co-expressing different markers from acetabular (Ac) and femoral head (Fh) bone samples of patients suffering from primary hip osteoarthritis (pOA) and secondary osteoarthritis due to developmental dysplasia of the hips (DDH-OA). Horizontal lines and boxes are median and IQR; statistical significance is shown on plots (*p* < 0.05, Mann–Whitney U test). Results that were statistically significant are bolded.

**Table 1 ijms-25-05173-t001:** Demographic and clinical characteristics of patients with primary hip osteoarthritis (pOA) and secondary hip osteoarthritis due to developmental dysplasia (DDH-OA).

	pOA	DDH-OA	*p*
Number of patients	12	12	n/a
Male/Female	4/8	1/11	0.1399 *
**Age [years]**	65.00 ± 7.87	50.58 ± 8.08	**0.0002 ****
BMI [kg/m^2^]	31.38 ± 7.16	29.03 ± 5.68	0.3833 **
Duration of pain [years]	3.0 [2.5–5.5]	3.0 [2.0–7.5]	0.9294 ***
VAS pain while resting	5.0 [3.0–7.0]	6.0 [5.5–8.0]	0.1766 ***
VAS pain in activity	8.5 [7.5–10.0]	9.5 [8.0–10.0]	0.3242 ***
Total WOMAC [%]	61.20 ± 10.40	58.25 ± 12.99	0.5457 **
mHHS	40.00 ± 8.94	40.17 ± 11.20	0.9682 **

Values are expressed as mean ± standard deviation or median [interquartile range]; Results that were statistically significant are bolded; * Chi-square test, ** *t*-test, *** Mann–Whitney test. Abbreviations: pOA = primary hip osteoarthritis, DDH-OA = secondary hip osteoarthritis due to developmental dysplasia, BMI = body mass index, VAS = visual analogue scale, WOMAC = Western Ontario and McMaster Universities Osteoarthritis Index, mHHS = modified Harris hip score.

**Table 2 ijms-25-05173-t002:** Intra-group comparison of bone area/tissue area (BA/TA) proportions from total, subchondral, and trabecular area from acetabular (Ac) and femoral head (Fh) bone samples of patients suffering from primary hip osteoarthritis (pOA) and secondary osteoarthritis due to developmental dysplasia of the hips (DDH-OA).

Group		Fh	Ac	*p **
pOA	**BA/TA**	38 [27.2–48.7]	47.4 [41.8–53.5]	**0.0342**
	**Subchondral BA/TA**	50.4 [33.3–60.5]	73.5 [63.4–78]	**0.0049**
	Trabecular BA/TA	38.4 [26.9–53.2]	46.4 [30–56]	0.9697
DDH-OA	BA/TA	37.6 [24.7–48.6]	35.7 [22.3–44.9]	0.4697
	Subchondral BA/TA	38.8 [28–70.7]	56.4 [40.8–74.1]	0.2036
	Trabecular BA/TA	44.2 [24.8–53]	33 [17.4–44.7]	0.1294

Values are expressed as median [interquartile range]; Results that were statistically significant are bolded; * Wilcoxon test. Abbreviations: Fh = femoral head, Ac = acetabulum, pOA = primary hip osteoarthritis, DDH-OA = secondary hip osteoarthritis due to developmental dysplasia, BA/TA = bone area/tissue area.

**Table 3 ijms-25-05173-t003:** Comparison of paired samples taken from patients suffering from primary hip osteoarthritis.

	Fh	Ac	*p **
CD10+	31.7 [23.2–37.8]	39.1 [18.9–45.2]	0.9097
CD140b+	27.4 [14.7–35.2]	25.3 [16.1–38.3]	0.6100
CD271+	51.5 [45.4–70.2]	48.1 [31.7–76.4]	0.4697
GD2+	8.1 [5.5–14.9]	12.5 [8.1–31]	0.0640
CD73+	21.9 [18.1–28.2]	18.2 [12.4–33.8]	0.9697
PDPN+	4.6 [1.9–8.6]	6.8 [4.4–16.6]	0.0771
CD146+	25 [14.1–37.9]	12.5 [6.8–32.1]	0.2036
CD164+	57.1 [36.5–62.3]	39.5 [25.6–51.5]	0.1466

Values are expressed as median [interquartile range]; * Wilcoxon test. Abbreviations: Fh = femoral head, Ac = acetabulum, CD = cluster of differentiation, GD2 = disialoganglioside, PDPN = podoplanin.

**Table 4 ijms-25-05173-t004:** Comparison of paired samples taken from patients suffering from secondary osteoarthritis due to developmental dysplasia of the hips.

	Fh	Ac	*p **
CD10+	19.1 [16.7–43.8]	18.1 [13.9–28.7]	0.0923
CD140b+	28.2 [13.2–38.1]	12.5 [7.3–34.7]	0.1763
**CD271+**	53.6 [30–86.7]	28.8 [18.8–41.9]	**0.0093**
GD2+	4.8 [2.5–7.2]	4.3 [1.2–19]	0.9097
CD73+	16.5 [11.4–28.4]	29.4 [12.6–40.4]	0.4238
PDPN+	10 [2.3–27]	5.3 [0.8–36.3]	0.7334
CD146+	32.4 [22.6–36.3]	22.5 [7.9–40.2]	0.0771
CD164+	67.4 [51.4–78.5]	65.5 [49.7–79.3]	0.7910

Values are expressed as median [interquartile range]; Results that were statistically significant are bolded; * Wilcoxon test. Abbreviations: Fh = femoral head, Ac = acetabulum, CD = cluster of differentiation, GD2 = disialoganglioside, PDPN = podoplanin

**Table 5 ijms-25-05173-t005:** Comparison of paired samples taken from patients suffering from primary hip osteoarthritis.

	Fh	Ac	*p **
CD10+CD271+	31.3 [29.3–42.8]	25.2 [11.7–37]	0.0771
**CD10+GD2+**	4.2 [1.3–7.9]	8.8 [6.1–23.4]	**0.0122**
**CD271+GD2+**	3.6 [2.7–6.6]	9.1 [5.2–19.8]	**0.0342**
**CD271+GD2+CD10+**	3.1 [1.2–5.3]	7.9 [3.9–16.5]	**0.0269**
CD164+CD146−	27.8 [9.6–32.3]	21.6 [10.3–28.8]	0.5301
CD164+CD146−CD73+	18.2 [7.9–25.4]	12.3 [7.6–20.8]	0.4697
**CD164+CD146**−**PDPN+**	2 [0.7–3.9]	3.4 [1.7–10.7]	**0.0469**
CD164+CD146−PDPN+CD73+	1.9 [0.8–3.9]	3.1 [1.6–10.7]	0.0829

Values are expressed as median [interquartile range]; Results that were statistically significant are bolded; * Wilcoxon test. Abbreviations: Fh = femoral head, Ac = acetabulum, CD = cluster of differentiation, GD2 = disialoganglioside, PDPN = podoplanin.

**Table 6 ijms-25-05173-t006:** Comparison of paired samples taken from patients suffering from secondary osteoarthritis due to developmental dysplasia of the hips.

	Fh	Ac	*p **
CD10+CD271+	15.9 [13.4–42.3]	13.6 [10.1–18.2]	0.0640
CD10+GD2+	2.9 [ 1.8–4.2]	3.2 [1–12.6]	0.5828
CD271+GD2+	3 [2–4.6]	2.4 [0.7–5.8]	0.9097
CD271+GD2+CD10+	2.2 [1.4–3.3]	1.6 [0.7–5.4]	0.7334
CD164+CD146−	31.2 [19.8–51.3]	40.4 [30.3–51.3]	0.1820
CD164+CD146−CD73+	15.2 [8.7–20.3]	26.2 [12–30.3]	0.2036
CD164+CD146−PDPN+	5.3 [1.3–12.9]	3.5 [0.8–20.7]	0.1971
CD164+CD146−PDPN+CD73+	3.4 [0.9–10.8]	3.1 [0.4–17.9]	0.2477

Values are expressed as median [interquartile range]; * Wilcoxon test. Abbreviations: Fh = femoral head, Ac = acetabulum, CD = cluster of differentiation, GD2 = disialoganglioside, PDPN = podoplanin.

## Data Availability

Most of the data are contained within the article or Appendix A. Additional data are available on request from the authors.

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
