# Peer review of "Distinctiveness of Femoral and Acetabular Mesenchymal Stem and Progenitor Populations in Patients with Primary and Secondary Hip Osteoarthritis Due to Developmental Dysplasia"

_ijms, 2024, doi:10.3390/ijms25105173_

Round 1

Reviewer 1 Report

Comments and Suggestions for Authors

The reviewer would like to aknowledge the extensive work accomplished by the authors. The reviewer suggests the following:

1. Introduction section can be further extended to include:

1.1. Current state of the art, regarding regenerative approache that rely on the use/modulation os bone marrow stem cells.

1.2. A more detailed rational for the chosen surface markers.

2. Results section - Figures 3., 4., 5., and 6. It would facilitate results´ visualization if the the control image appear first in the panels. 

3. Results section - it would be easier for the readers if the figures were crossreferenced in the body of text.

4. Discussion section could be extended to deepen how the study findings woul impact regenerative approches to osteoarthritis.

5. Aknowledge in the discussion section that the histomorphometrical characterization does not account for mineralization quantity/quality.

Reviewer 2 Report

Comments and Suggestions for Authors

The manuscript ijms-2987884, "Distinctiveness of femoral and acetabular mesenchymal stem and progenitor populations in patients with primary and secondary hip osteoarthritis due to developmental dysplasia", reports the differences of primary hip OA (n=12) and secondary pOA (DDH-OA; n=12) in subchondral morphology and proportions of non-hematopoietic cells expressing MSPC markers were noted depending on OA type and skeletal location. The limitation of many current OA clinical studies is the lack of healthy control. 

However, the insight from comparing the primary and secondary OA, identifies cells in subchondral bone expressing common MSPC markers in vivo and compares the proportions of these populations in end-stage pOA vs DDH-OA as well as correlates them with the clinical, demographic, and morphological characteristics of the two groups are novel and important to the field. The authors also looked at the acetabular and femoral head bone in hip OA. The result suggested bone sclerosis was more prominent in pOA acetabulum (Ac) in comparison to DDH-OA Ac, and pOA Ac compared to pOA femoral head (Fh).

The manuscript aimed to submit to the topic collection "Osteoarthritis: From Molecular Mechanism to Novel Therapy" which is relevant. It can be published after addressing the following questions.

The authors mentioned the lack of understanding of OA pathogenesis, "Osteoarthritis (OA) is a progressive degenerative joint disease whose pathogenesis is still not completely understood [1,2]" (line 39). 

To my knowledge, the development of primary OA is not caused by nothing. There are lots of risk factors that have been reported associated with primary OA, such as aging, and biological and physical stressors. Genomics, epigenetics, and metabolism have provided lots of evidence that could explain the hemostats changes in the cells for example of chondrocytes. That would be appreciated to provide more details about the OA before leading to the MSPC. The authors focused on the difference between primary hip OA vs secondary hip OA. That would also be appreciated to briefly introduce the known difference between OA in different joints.

In the section on Morphological characteristics of femoral and acetabular samples, no significant differences were identified between pOA and DDH-OA, however, there is an obvious trend of BA/TA in the acetabular bone. Did the authors quantify the thickness of subchondral bone or other parameters?

It would be also great to provide the representative CT scan image results to support the difference you see from the Ac. Can the author provide any prediction on the high bone mass identified in the pOA?

Can the authors prove that the difference observed, i.e. MSPC populations,  in DDH-OA and pOA patients, is not the difference comparing the young and old individuals?

The tSNE plot showed some subclusters. Are you able to identify the cell types? Does the tSNE plot represent a merge of all samples or one sample? It would be appreciated if you could add a color bar indicating the intensity or mention it in the figure legend. Also, can you label the multiple markers at the same time in the tSNE to show the subpopulation difference, that will be more instructive.

In the discussion, would it be better to put the paragraph about BA/TV up (line 363), and keep the same flow as your main text?

Reviewer 3 Report

Comments and Suggestions for Authors

This article shows the distinctiveness of femoral and acetabular mesenchymal stem and progenitor populations in patients with primary and secondary hip osteoarthritis due to developmental dysplasia. The topic is relevant, but the major deficiencies identified in both content and form need to be addressed based on the specific recommendations below:

1. The concluding part of the abstract should be improved in terms of results and future research directions to which this research can refer.

2. No blank spaces between paragraphs are required.

3. As the introduction is too poorly addressed in relation to the complexity of the topic, a more detailed approach to OA management is needed. I suggest checking and referring to: PMID: 34868573 and PMID: 35454333.

4.  The aim of the paper must be addressed also from the perspective of describing the contribution to the field under analysis and the elements of scientific novelty presented.

5. Given the complexity of the subject and the results obtained, it is advisable to create a separate conclusions section at the end, highlighting the most important aspects of the work together with future research studies.

Round 2

Reviewer 2 Report

Comments and Suggestions for Authors

The authors have addressed my concerns.

Reviewer 3 Report

Comments and Suggestions for Authors

The authors have significantly improved the manuscript based on the suggestions received.